# Two Novel Genera, *Neostemphylium* and *Scleromyces* (*Pleosporaceae*) from Freshwater Sediments and Their Global Biogeography

**DOI:** 10.3390/jof8080868

**Published:** 2022-08-17

**Authors:** Daniel Torres-Garcia, Dania García, José F. Cano-Lira, Josepa Gené

**Affiliations:** Unitat de Micologia i Microbiologia Ambiental, Facultat de Medicina i Ciències de la Salut and IISPV, Universitat Rovira i Virgili, 43201 Reus, Spain

**Keywords:** *Ascomycota*, geographic distribution, GlobalFungi, Phylogeny, *Pleosporales*, taxonomy

## Abstract

Although the *Pleosporaceae* is one of the species-richest families in the *Pleosporales*, research into less-explored substrates can contribute to widening the knowledge of its diversity. In our ongoing survey on culturable *Ascomycota* from freshwater sediments in Spain, several pleosporacean specimens of taxonomic interest were isolated. Phylogenetic analyses based on five gene markers (ITS, LSU, *gapdh*, *rbp*2, and *tef*1) revealed that these fungi represent so far undescribed lineages, which are proposed as two novel genera in the family, i.e., *Neostemphylium* typified by *Neostemphylium polymorphum* sp. nov., and *Scleromyces* to accommodate *Scleromyces submersus* sp. nov. *Neostemphylium* is characterized by the production of phaeodictyospores from apically swollen and darkened conidiogenous cells, the presence of a synanamorph that consists of cylindrical and brown phragmoconidia growing terminally or laterally on hyphae, and by the ability to produce secondary conidia by a microconidiation cycle. *Scleromyces* is placed phylogenetically distant to any genera in the family and only produces sclerotium-like structures in vitro. The geographic distribution and ecology of *N. polymorphum* and *Sc. submersus* were inferred from metabarcoding data using the GlobalFungi database. The results suggest that *N. polymorphum* is a globally distributed fungus represented by environmental sequences originating primarily from soil samples collected in Australia, Europe, and the USA, whereas *Sc. submersus* is a less common species that has only been found associated with one environmental sequence from an Australian soil sample. The phylogenetic analyses of the environmental ITS1 and ITS2 sequences revealed at least four dark taxa that might be related to *Neostemphylium* and *Scleromyces*. The phylogeny presented here allows us to resolve the taxonomy of the genus *Asteromyces* as a member of the *Pleosporaceae*.

## 1. Introduction

The *Pleosporaceae* is one of the largest families within the order *Pleosporales* (*Dothideomycetes*) in terms of the number of species. It was introduced by Nitschke [1] and was considered for a long time a heterogeneous group of bitunicate ascomycetes with its genera distinguished primarily by their ascospore features (i.e., shape, color, septation, pigmentation, and presence or lack of mucilaginous sheaths) [2]. According to recent taxonomic revisions of the *Dothideomycetes* [3,4], based on morphological investigations and phylogenetic data, the *Pleosporaceae* is a well-delineated family that comprises 23 genera and more than 2000 species. *Alternaria*, *Bipolaris*, *Curvularia*, *Exserohilum*, *Pyrenophora*, and *Stemphylium* are the most species-rich genera in the family [3]. They show teleomorphs characterized by black ostiolate ascomata with thick-walled peridium, cellular pseudoparaphyses and bitunicate, fissitunicate, eight-spored asci, producing melanized, phragmosporous, or muriform ascospores [3]. More commonly, they present dematiaceous hyphomycetous anamorphs producing phragmo- or dyctioconidia from tretric (poroblastic) or blastic conidiogenous cells, although coelomycetous anamorphs with phialidic or anellydic conidia have also been described [3,5]. Although *Pleospora* was designed as the type genus in the family, with the advent of one fungus-one name initiative in the International Code of Nomenclature for algae, fungi, and plants (ICN; Melbourne Code) [6], the name *Stemphylium* was retained over *Pleospora* by the working group on *Dothideomycetes* of the International Commission on the Taxonomy of Fungi [7].

Members of the *Pleosporaceae* are widely distributed across the environment and have a wide range of lifestyles, i.e., saprophytic, endo-/epiphytic, and parasitic on various hosts in terrestrial and aquatic environments [8]. Among them, species of *Alternaria*, *Bipolaris*, *Curvularia*, or *Stemphylium* are important pathogenic fungi to plants of various crops, resulting in yield and economic losses [8,9,10]. However, they also include human and animal pathogens that cause infections with different clinical manifestations [11]. Several metagenomic studies reveal that pleosporalean fungi are well represented in aquatic environments [12,13,14]. Although some species have been found strictly adapted to aquatic ecosystems [15], many of them are commonly found in association with terrestrial plants and, therefore, they are not considered especially adapted to freshwater habitats [16]. Results of those studies also suggest that this group of fungi has been generally overlooked and undersampled in freshwater ecosystems, particularly in rivers, despite their relevant role in ecosystem functioning as saprophytes and parasites [12,17].

In our latest efforts to expand knowledge on the diversity of culturable *Ascomycota* from river sediments collected in Spain, several interesting specimens of dematiaceous filamentous fungi were isolated. A preliminary sequence analysis of the nuclear ribosomal operon (i.e., the 28S large ribosomal subunit––LSU, and the internal transcribed spacer––ITS, including the 5.8S rDNA gene) revealed that those specimens would belong to the *Pleosporaceae*, but they could not be identified at the genus level. The aim of the present study was, therefore, to resolve the taxonomy of the above-mentioned isolates based on morphological features and multi-locus phylogenetic analysis inferred with sequences of the nuclear markers mostly represented in the different members of the *Pleosporaceae*. These are the LSU and ITS regions of the rDNA, and partial fragments of the RNA polymerase II largest subunit (*rpb*2), the translation elongation factor 1–α (*tef*1), and the glyceraldehyde-3-phosphate dehydrogenase (*gapdh*) genes [8,18,19]. Additionally, in order to elucidate the putative global geographic distribution of those isolates and to study their diversity hidden among environmental sequences, their ITS barcodes (i.e., full-length of ITS1 and ITS2 sequences) were blasted against the GlobalFungi database [20]. This is a recently created database, which currently includes accumulated data on fungal distribution and ecology generated from more than three hundred metagenomic studies published in the last decade (GlobalFungi database, accessed on 18 May 2022).

## 2. Materials and Methods

### 2.1. Sampling and Isolates

Sediment samples were collected in 2019 from natural areas of two Spanish provinces, Lleida and Madrid. Samples from Lleida were collected from the Segre River as it passes through Camarasa, an area characterized by a continental Mediterranean climate (https://www.meteo.cat/wpweb/climatologia/el-clima-ahir/el-clima-de-catalunya/, accessed on 21 April 2022), with an average annual temperature of 13.5 °C, an average annual rainfall of 800 mm, an altitude of 800 m, and a vegetation dominated by holm oaks (*Quercus ilex* subsp. *rotundifolia*) (http://www.biodiver.bio.ub.es, accessed on 21 April 2022). Samples from Madrid were collected from two streams around Rascafría in the Guadarrama Natural Park. This area has a continental mountain climate with an average annual temperature of 11.8 °C, an average annual rainfall of 530 mm, an altitude of 1200 m, and a forest dominated by *Cistus oromediterraneus*, *Juniperus communis,* and *Pinus sylvestris* (https://www.parquenacionalsierra guadarrama.es, accessed on 21 April 2022).

Sediments from the rivers or streams selected in the above-mentioned locations were collected randomly. Samples were obtained ca 10 cm below the surface layer from the riverbeds or edges using sterile 100 mL plastic containers, which were transported in a refrigerated container to the laboratory and processed immediately. Samples were vigorously shacked in the same containers; then, after 1 min at rest, the water was decanted and the sediment was poured into plastic trays onto several layers of sterile filter paper to remove excess water [21]. To achieve a greater fungal diversity in culture, three agar media were used: dichloran rose-bengal-chloramphenicol agar (DRBC; 2.5 g peptone, 5 g glucose, 0.5 g KH_2_PO_4_, 0.25 g MgSO_4_, 12.5 mg rose-bengal, 100 mg chloramphenicol, 1 mg dichloran, 10 g agar, 500 mL distilled water), DRBC supplemented with 0.01 g/L of benomyl, and potato dextrose agar (PDA; Pronadisa) supplemented with 2 g/L of chloramphenicol and 2 g/L of cycloheximide. Each sample was cultured in duplicate in each medium as follow: 0.5 g of sediment was mixed with melted medium at 45 °C in the same Petri dish and, once solidified, it was incubated at room temperature (22–25 °C) in the dark. Plates were examined weekly by stereomicroscope for 4–5 weeks. To obtain pure cultures, fragments of the colony or conidia of the fungi growing on primary cultures were transferred, using a sterile dissection needle, to plates containing PDA supplemented with chloramphenicol and incubated at 25 °C in darkness. These PDA cultures were used for a preliminary morphological identification and for extracting DNA of the fungi selected.

Living cultures of putative novel or rare fungi were preserved and deposited in the culture collection of the Faculty of Medicine in Reus (FMR, Spain) for further studies. Taxonomic information and nomenclature for the new species were deposited in MycoBank (https://www.mycobank.org/, accessed on 23 March 2022). Cultures from ex-type strains and holotypes, which consisted of dry colonies on the most appropriate media for their sporulation, were also deposited at the Westerdijk Fungal Biodiversity Institute in Utrecht (CBS, The Netherlands) (https://wi.knaw.nl/, accessed on 22 May 2022).

In addition, the ex-type and a reference strain of *Asteromyces cruciatus* were also examined in the current study, because a preliminary molecular comparison revealed this species as related to some of our isolates. According to Mycobank and the Index Fungorum database, *A. cruciatus* represents a monotypic genus with unclarified taxonomy.

### 2.2. Phenotypic Study

Microscopic characterization was carried out from the isolates growing on potato carrot agar (PCA; 20 g potato, 20 g carrot, 13 g agar, 1 L distilled water) after 7–14 d at 25 °C in darkness and mounted on slides with Shear’s mounting solution (3 g potassium acetate, 60 mL glycerol, 90 mL ethanol 95%, and 150 mL distilled water) [22], using an Olympus BH-2 bright field microscope (Olympus Corporation, Tokyo, Japan). Size ranges of relevant structures in species descriptions were derived from at least 30 measurements. Micrographs were taken using a Zeiss Axio-Imager M1 light microscope (Zeiss, Oberkochen, Germany) with a DeltaPix Infinity × digital camera. Photoplates were assembled from separate photographs using PhotoShop CS6. Macroscopic characterization of the colonies was made on PDA, PCA and oatmeal agar (OA; 30 g oatmeal, 13 g agar, 1 L distilled water) after 7 days at 25 °C in darkness. Other culture media, such as OA and PCA with sterile plant debris (i.e., leaves and twigs of *Dianthus caryophyllus*), synthetic nutrient-poor agar (SNA; 1 g KH_2_PO_4_, 1 g KNO_3_, 0.5 g MgSO_4_ × 7H_2_O, 0.5 g KCl, 0.2 g glucose, 0.2 g sucrose, 14 g agar, 1 L of distilled water), and V8 medium (16 g agar, 200 mL V8 juice, 1 L distilled water), were also used specifically for the FMR 18289 in order to stimulate its sporulation. For the same purpose, this isolate was submitted to the procedure described in Nishikawa and Nakashima [23] for *Alternaria* sporulation. Color notations in descriptions were according to Kornerup and Wanscher [24]. Growth rates were measured in duplicate on PDA after 7 d in darkness, at 5 °C intervals from 5 to 40 °C, and also at 37 °C.

### 2.3. DNA Extraction, Sequencing and Phylogenetic Analysis

Total genomic DNA was extracted through the modified protocol of Müller et al. [25] and quantified using Nanodrop 2000 (Thermo Scientific, Madrid, Spain). In order to reconstruct the phylogeny of the *Pleosporaceae* family, the *loci* amplified and sequenced were the ITS barcode and the D1/D2 domains of the LSU of the rDNA, as well as gene fragments of the *rpb*2, *tef*1, and *gapdh*. Primers pairs for their amplification were ITS5/ITS4 [26], LR0R/LR5 [27], RPB2-5F2/fRPB2-7cR [28,29], EF1-728F/EF1-986R [30], and gpd1/gpd2 [31], respectively. Briefly, PCR conditions for ITS, LSU, *gapdh,* and *tef*1 were set as follows: an initial denaturation at 95 °C for 5 min, followed by 35 cycles of 30 s at 95 °C, 45 s at 56 °C, and 1 min at 72 °C, and a final extension step at 72 °C for 10 min. PCR conditions for the *rpb*2 were an initial denaturation at 94 °C for 5 min, followed by 5 cycles of 45 s at 94 °C, 45 s at 60 °C, and 2 min at 72 °C, then 5 cycles of 45 s at 94 °C, 45 s with 58 °C, and 2 min at 72 °C, later 30 cycles of 45 s at 95 °C, 45 s with 54 °C, and 2 min at 72 °C, and a final extension step at 72 °C for 7 min. PCR products were purified and sequenced at Macrogen Corp. Europe (Madrid, Spain) with the same primers used for amplification. Consensus sequences were assembled using SeqMan v. 7.0.0 (DNAStar Lasergene, Madison, WI, USA).

A preliminary species identification of the sediment isolates was carried out by comparing their ITS region with those at the National Center for Biotechnology Information (NCBI) using the Basic Local Alignment Search Tool (BLAST; https://blast.ncbi.nlm. nih.gov/Blast.cgi, accessed on 5 March 2021) and with the UNITE database (https://unite.ut.ee/, accessed on 5 March 2021). A maximum similarity level of ≥98% was used for species-level identification. Lower similarity values were considered as putative unknown fungi, and their taxonomic position was assessed from analyses of the *loci* mentioned above.

Sequences of related species and representatives of other genera belonging to the *Pleosporaceae* family were obtained from GenBank and are listed in Table 1.

Individual and combined analyses using the LSU, ITS, *gapdh*, *rpb*2, and *tef*1 sequences were carried out to assess the phylogenetic relationship of the unidentified isolates to the other taxa in the family. Datasets for each locus were aligned individually in MEGA (Molecular Evolutionary Genetics Analysis) software v.6.0 [46], using the ClustalW algorithm [47] and refined with MUSCLE [48] or manually adjusted, if necessary, on the same platform. Phylogenetic concordance of the five-locus datasets was tested individually in each single-locus phylogeny through visual comparison and using the Incongruence Length Difference (ILD) implemented in the Winclada program [49] in order to assess any incongruent results among nodes with high statistical support. Once their concordance was confirmed, individual alignments were concatenated into a single data matrix with SequenceMatrix [50]. The best substitution model for all gene matrices was estimated using MEGA software for Maximum Likelihood (ML) analysis, whereas for the Bayesian Inference (BI) analysis it was estimated using jModelTest v.2.1.3 following the Akaike criterion [51,52]. The phylogenetic reconstructions were performed with the combined genes using ML under RAxML-HPC2 on XSEDE v-8.2.12 [53] in CIPRES Science gateway portal [54] and BI with MrBayes v.3.2.6 [55].

For the ML analysis, phylogenetic support for internal branches was assessed by 1000 ML bootstrapped pseudoreplicates and bootstrap support (bs) ≥ 70 was considered significant [56]. The phylogenetic reconstruction by BI was carried out using 5 million Markov Chain Monte Carlo (MCMC) generations, with four runs (one cold chain and three heated chains), and samples were stored every 1000 generations. The 50% majority-rule consensus tree and posterior probability (pp) values were calculated after discarding the first 25% of samples. A pp value of ≥0.95 was considered significant [57]. The resulting trees were plotted using FigTree v.1.3.1 (http://tree.bio.ed.ac.uk/software/figtree/, accessed on 1 June 2022). The DNA sequences and alignments generated in this study were, respectively, deposited in GenBank (Table 1) and in Zenodo (https://doi.org/10.5281/zenodo.6973696, accessed on 9 August 2022). 

### 2.4. Phylogeny and Geographic Distribution of Allied Environmental Sequences

In order to assess putative global geographic distribution and ecology of the novel fungi detected and their hidden diversity among environmental sequences, the full length of their ITS1 and ITS2 sequences were blasted against the GlobalFungi database [20]. At the time of accession (May 2022), this dataset contained 36,684 samples from 367 studies, 213, 747, 241 unique sequences for ITS1, and 582, 264, 149 for ITS2. Since GlobalFungi has separated ITS1 and ITS2 sequences, they were analyzed separately. In order to verify generic and species boundaries among downloaded ITS environmental sequences related to our fungi, we also included in the analyses ITS sequences of known species previously obtained from the GenBank and UNITE databases. Those known species were representatives of the well-delineated monophyletic genera (i.e., *Alternaria*, *Asteromyces, Gibbago*, *Paradendryphyella*, *Pyrenophora,* and *Stemphylium*), which were the closest taxa to our fungi in a full length ITS analysis carried out previously (Appendix A). ITS1 and ITS2 sequences of some of those fungi were also blasted against the GlobalFungi dataset and included in the respective analyses. In each case, we selected and downloaded environmental sequences that had a similarity of between 98 and 100% and a full-length coverage with the sediment isolates and with those related to the known species, apart from the ITS1 of the isolate FMR 18289, because its highest sequence similarity found in the database was lower, at 95%. Pleosporacean genus/species boundaries were inferred from ML trees of ITS1 and ITS2 sequences computed in RAxML. Virtual taxa, consisting of environmental sequences only, were defined as arbitrary phylotypes in the phylogenetic trees, following Réblová et al. [58,59]. Data on occurrence across environmental samples and metadata related to the particular samples (location, substrate, biome, or climatic data) were obtained for each taxon and are listed in Appendix A.

## 3. Results

Among pleosporacean fungi found in the freshwater sediments, we recovered five isolates (FMR 17886, FMR 17889, FMR 17893, FMR 17894, FMR 17895) exclusively from DRBC agar supplemented with benomyl. These isolates were identified initially as *Stemphylium* sp. because they showed similar morphological features but did not exactly fit into any of the known species described in that genus. Another interesting isolate (FMR 18289) was recovered from DRBC but could not be identified morphologically because it only produced sclerotium-like structures and failed to form fertile reproductive morphs (i.e., anamorph and/or teleomorph) despite the attempts to stimulate sporulation in various culture conditions.

### 3.1. Phylogeny

Molecular identification based on the BLAST query revealed that LSU sequences of the six unidentified isolates showed a high percentage of similarity with other members of the *Pleosporaceae*. Specifically, the stemphylium-like isolates showed a sequence identity of 99% with *Stemphylium* (*S.*) *vesicarium* (CBS 191.86) and *Bipolaris* (*B.*) *microlaenae* (CBS 280.91), while the sequence of the sclerotium-forming isolate was 99% similar to *Pyrenophora* (*P*.) *seminiperda* (CBS 127927) and 98% to *Alternaria* (*A.*) *avenicola* (CBS 121459). Similar values were obtained when sequences of species of other well-delineated genera in the *Pleosporaceae* were compared, which confirmed the low discriminatory power of this gene marker in the family. On the other hand, the genetic similarity was considerably lower when ITS sequences were compared with other members of the *Pleosporaceae*. The closest matches for the stemphylium-like isolates were *Paradendryphiella* (*Pa.*) *salina* (CBS 142.60 and CBS 141.60) with a similarity of 96%, followed by *Pa. areniae* (CBS 181.58) and *S. vesicarium* (CBS 191.86) with a 95%. BLAST results and the particular morphology of those isolates precluded them from being classified in the genus *Stemphylium* or in *Paradendryphiella*. The highest similarity for ITS sequence of the remaining isolate (FMR 18289) was 96% with *A. avenicola* (CBS 121459), followed by *P. seminiperda* (CBS 127927) with a similarity of 90%. BLAST searches using the remaining phylogenetic markers revealed even lower values of similarity (≤89.6%) with other members of the *Pleosporaceae*.

Since individual analyses with LSU, ITS, *gapdh*, *rpb*2, and *tef*1 were visually similar and the ILD test did not show incongruences (*p* = 0.33), a multi-gene analysis was carried out with the five markers. The concatenated phylogeny encompassed 59 sequences that represented 17 genera in the *Pleosporaceae* with 3160 bp long (531 for ITS, 892 for LSU, 865 for *rpb*2, 624 for *gapdh*, 248 for *tef*1), of which 1164 were variable sites (198 for ITS, 168 for LSU, 375 for *rpb*2, 281 for *gapdh*, 142 for *tef*1) and 905 were phylogenetically informative sites (159 for ITS, 94 for LSU, 325 for *rpb*2, 231 for *gapdh*, 96 for *tef*1). For the ML analyses, K2 + G + I was selected as the best fit model for ITS, LSU, and *rpb*2, the K2 + G for *tef*1 and TN93 + G for *gapdh*. For the BI analyses, SYM + G + I was selected as the best fit model for ITS and *rpb*2, the K2 + G + I for LSU, the HKY + G + I for *gapdh*, and the K2 + G for *tef*1. The RAxML tree (Figure 1) showed that FMR 17886, FMR 17889, FMR 17893, FMR 17894, and FMR 17895 clustered together in a monophyletic undescribed lineage, strongly supported (100 bs/1 pp), which was sister to a well-supported clade (83 bs/0.98 pp) that includes members of the genera *Asteromyces*, *Paradendryphiella,* and *Stemphylium*. Among these, *Asteromyces* currently has an uncertain taxonomic position among the *Ascomycota*. However, our analysis places its type species, *A. cruciatus*, into the *Pleosporaceae*. The undescribed lineage of the five sediment isolates represents a novel genus, which is proposed here as *Neostemphylium* (*N.*) and represented by the new species *N. polymorphum* (Figure 1). These isolates showed similar morphological features and had an intra-specific genetic variability ranging from 0.1 to 0.4% in the concatenated phylogenetic analysis. In the same phylogeny (Figure 1), the isolate FMR 18289 was located in a faraway single branch within a well-supported clade (99 bs/1 pp) together with other genera of the *Pleosporaceae*, representing a novel genus for the family. This is proposed below as *Scleromyces* (*Sc.*) and typified by the new species *Sc. submersus.* A detailed morphological characterization of the novel fungi is provided in the taxonomy section.

### 3.2. Biogeography and Ecology

A BLAST search in the GlobalFungi database revealed the presence of *Neostemphylium* and *Scleromyces* among environmental sequences from samples collected worldwide. When we compared the ITS1 sequences of *Neostemphylium*, this resulted in 469 unique environmental ITS1 sequences (similarity 98–100%), covering 739 samples. The ITS1 sequence of the *Scleromyces* isolate, as mentioned before, yield the highest similarity value, that of 95%, found in the database .At that value, we obtained 500 environmental sequences, which covered 291 samples. With so many environmental sequences related to our fungi, we were able to select for the analyses representatives from a variety of locations, substrates, and biomes (Appendix A), as well as other environmental sequences from different species of the genera *Alternaria, Pyrenophora,* and *Stemphylium*. The ITS1 phylogenetic analysis included 102 sequences, 225 characters, of which 156 were variable sites and 127 were phylogenetically informative sites. The ML tree was rooted in a branch leading to *Comoclathris* (*Co.*) *typhicola* (CBS 132.69) and *Co. sedis* (CBS 366.52) (Figure 2). The environmental ITS1 sequences selected clustered into eight phylotypes, two of which, with a total of 32 sequences, were related to the genus *Neostemphylium*, and another two phylotypes with three sequences were related to *Scleromyces*. 

Most environmental sequences linked to *Neostemphylium* formed the phylotype representative of *N. polymorphum*, apart from four sequences that were designated as the phylotype ITS1-ENV1 and could represent a hypothetically distinct species from *N. polymorphum.* On the other hand, no environmental ITS1 sequences were matched to the novel species *Sc. submersus*. However, this species formed a divergent branch close to the phylotypes designated as ITS1-ENV2 and ITS1-ENV3, with one and two sequences, respectively, which might also represent two hidden taxa for the genus *Scleromyces*.

In contrast, more environmental ITS2 sequences were found related to the genus *Scleromyces* than to *Neostemphylium*. Namely, 26 sequences 98–100% similar were linked to *Scleromyces* and eight unique sequences to the latter genus, covering six and nine samples, respectively. The ITS2 dereplicated dataset had 92 sequences that were representative of members of the above-mentioned genera, with 166 characters, of which 90 were variable sites and 73 phylogenetically informative sites. The ML tree was rooted to *Co. typhicola* (CBS 132.69) and *Co. sedis* (CBS 366.52) (Figure 2). The environmental sequences were distributed into seven phylotypes. Specifically, eight sequences were linked to the phylotype of *N. polymorphum* and one to *Sc. submersus*, while the remaining sequences related to *Scleromyces* were designated as phylotype ITS2-ENV1 because they represented a distinct species from *Sc. submersus*. However, this *Scleromyces* phylotype does not correlate with any delineated in the ITS1 analysis, since ITS2-ENV1 includes environmental sequences from the USA, while ITS1-ENV2 and ITS1-ENV3 phylotypes both have sequences from Australian samples.

Biogeography and ecological parameters of the environmental sequences related to our novel fungi and inferred in ITS1 and ITS2 phylogenetic analyses are summarized in Table 2. Briefly, Oceania (mainly Australia) has the majority of samples containing ITS1 and ITS2 sequences linked to *N. polymorphum*, *Sc. submersus,* and the hidden phylotypes identified here. Nevertheless, many sequences linked to *N. polymorphum* were also found in samples from areas of Europe (France and Spain) and the USA, the most-sampled areas in GlobalFungi (33.55% and 23.74%, respectively). Interestingly, the *Scleromyces* phylotype ITS2-ENV1 is the only sequence sampled from aquatic environments collected in the USA (Table 2). Conversely, sequences linked to *N. polymorphum*, *Sc. submersus,* and to the other phylotypes related to the novel genera were sampled from soils or roots as the most frequently inhabited substrates in different biomes (grasslands, wetlands, croplands, woodlands, shrublands, or, rarely, forests) (Table 2).

### 3.3. Taxonomy

***Neostemphylium*** Torres-Garcia, Gené and Cano, gen. nov.

MycoBank MB 843270

*Etymology*: Name refers to the morphological resemblance with *Stemphylium*.

Subclass classification: Dothideomycetes, Pleosporomycetidae, Pleosporales, Pleosporaceae.

*Type species*: *Neostemphylium polymorphum* Torres-Garcia, Gené and Cano.

*Description*: *Teleomorph* not observed. *Anamorph* hyphomycetous. *Conidiophores* semi-macronematous or macronematous, mononematous, consisting in conidiogenous cells growing terminally or laterally often on short supporting cells from vegetative hyphae, or septate and branched towards the upper part, branches sometimes with percurrent proliferations, subhyaline to pale brown, smooth-walled or verruculose mainly towards the upper part. *Conidiogenous cells* mono- or polyblastic, integrated or discrete, terminal or lateral, sometimes intercalary, subglobose, subcylindrical, barrel-shaped or obclavate, pale brown, subhyaline around the conidiogenous locus, smooth-walled to verruculose. *Conidia* dry, acropleurogenous, solitary or in short acropetal chains, subglobose, ellipsoidal, or oblong, muriform, often constricted at the transversal septa, brown to dark brown, finely roughened to verrucose, thick-walled. *Microconidiation* cycle from muriform primary conidia can be present. *Synanamorph* state can be present, consisting in blastic fragmoconidia, growing lateral or terminal on vegetative hyphae, sessile or short stalked, cylindrical, sometimes branched, rounded apically, truncate at base, pale brown to brown, thick-walled, often remaining attached on hyphae.

*Habitat and geographical distribution*: In addition to our freshwater sediment isolates from Spain, the environmental data suggest that members of *Neostemphylium* would primarily inhabit soil, but also air, rhizosphere soils, roots, and shoots from areas of Australia, Europe (France), and the USA (Figure 2).

***Neostemphylium polymorphum*** Torres-Garcia, Gené and Cano, sp. nov. Figure 3.

Mycobank MB 843271

*Etymology*: Name refers to the different structures produced by the fungus.

*Type*: Spain, Comunidad de Madrid, Rascafría, Arroyo del Brezal, 40°51′30.7″ N 3°54′37.6″ W, fluvial sediments, May 2019, *J. Cano* (**holotype** CBS H-24943; culture ex-type FMR 17886, CBS 149061).

*Description*: *Mycelium* immersed and superficial. *Hyphae* septate, branched, 2.5–4.5 μm wide, hyaline to pale brown, smooth-walled to verruculose. *Conidiophores* semi-macronematous or macronematous; semi-macronematous conidiophores consisting in conidiogenous cells arising terminally or often laterally from supporting cells or directly on hyphae, pale brown, verruculose; macronematous conidiophores straight to flexuous, septate, branched, up to 280 μm long, hyaline and smooth-walled towards the base, pale brown and mostly verruculose towards the branched part, with branches bearing terminally 1–2 conidiogenous cells, sometimes proliferating percurrently forming nodulose branches. *Conidiogenous cells* integrate and terminal or discrete growing laterally from the hyphae or conidiophore branches, sometimes intercalary due to the proliferation of the branches, subcylindrical or barrel-shaped, more commonly obclavate, 6–14.5 (–17.5) × 4.5–7 μm, pale brown to brown, usually verruculose. *Conidia* solitary, occasionally in short acropetal chains with up to 3 conidia, brown to dark brown, verruculose to verrucose, subglobose, ellipsoidal or oblong, (14–)18–26 (–29) × 11–16 μm, with 1–2 longitudinal or oblique septa per transversal segment and (1–)3–4 transverse septa often constricted, with a narrow cylindrical basal hylum. *Microconidiation cycle* was observed in some primary conidia, which give rise to secondary conidia, similar to the initial ones, from conidiogenous *loci* in the apical, lateral, or subterminal cells of the primary conidial body. *Synanamorph* present, consisting in blastic phragmoconidia, sessile or short stalked, with (1–)3–5(–7) septa, often constricted at septa, cylindrical to subcylindrical, sometimes branched, rounded apically, truncate at base, (20–)26–32(–36) × 7–10 μm, smooth- and thick-walled, brown, often remaining attached on hyphae.

*Culture characteristics* (7d at 25 °C): Colonies on PDA reaching 79–80 mm diam., flattened, dense, granulose, aerial mycelium scarce, sporulation abundant, dark green (30F8), yellowish green (30B7) at periphery, margins fimbriate and slightly irregular; reverse dark green (30F8) at center to golden brown (5D7) towards periphery, with a light yellow (4A4) soluble pigment. On PCA, attaining 61–62 mm diam., flattened, granulose, aerial mycelium scarce, sporulation abundant, dark green (29F5), greenish grey (28B2) at periphery, margins fimbriate and regular; reverse dull green (29E4), grey (28A2) at periphery, soluble pigment absent. On OA, reaching 52–54 mm diam., flattened, granulose, aerial mycelium scarce, sporulation abundant, dark green (28F6) at center, greenish grey (28B2) at periphery, margins fimbriate and regular; reverse dark green (28F4) to dull green (28E4) towards periphery, soluble pigment absent.

*Cardinal temperatures for growth*: minimum 5 °C, optimum 25 °C, maximum 35 °C.

*Additional isolates examined*: Spain, Comunidad de Madrid, Rascafría, Arroyo de la Umbría, 40°51′54.7″ N 3°53′40.3″ W, fluvial sediments, May 2019, *J. Cano* (FMR 17893, CBS 149062); Arroyo del Brezal, 40°51′31.5″ N 3°54′38.6″ W, fluvial sediments, May 2019, *J. Cano* (FMR 17889); Arroyo de la Umbría, 40°51′54.7″ N 3°53′40.3″ W, fluvial sediments, May 2019, *J. Cano* (FMR 17894); Arroyo de la Umbría, 40°51′38.6″ N 3°54′13.7″ W, fluvial sediments, May 2019, *J. Cano* (FMR 17895).

*Distribution*: Australia, France, Spain, and the USA (Figure 2, Table 2).

*Notes*: The multi-gene phylogeny of the *Pleosporaceae* presented here shows that *N. polymorphum* is related to the genera *Asteromyces*, *Paradendryphiella,* and *Stemphylium* (Figure 1). However, it is not only placed distant from the clade representative of these three genera, but *Neostemphylium* also differs in several diagnostic morphological features. Although *Neostemphylium* and *Stemphylium* resemble each other in their anamorphs characterized by the formation of phaeodictyospores from apically swollen conidiogenous cells, the conidiophore branching pattern is more complex in *Neostemphylium* than in *Stemphylium* species. Conidiophores in the latter genus are commonly unbranched or rarely branched [9]. In addition, *Neostemphylium* produces a synanamorphic state characterized by blastic, brown phragmoconidia, sometimes branched, that are not reported in any species of *Stemphylium*. *Paradendryphiella* differs in the production of exclusively cylindrical to obclavate phragmoconidia with dark septa on narrow denticles, often aggregated at the apex of lateral or terminal conidiogenous cells [18], while the conidiogenous apparatus of *Asteromyces* is characterized by polyblastic swollen conidiogenous cells with long denticles in a radial arrangement, giving rise to one-celled dark brown conidia [60].

*Gibbago* is another pleosporacean genus, represented by *G. trianthemae*, which also resembles *Neostemphylium* in its conidiogenous cells and conidial morphology [38,61]. However, like in *Stemphylium* species, *Gibbago* produces mostly unbranched or rarely branched conidiophores, and no synanamorph or microconidiation cycle have been described in *G. trianthemae*. In addition, our phylogeny agrees with the *Pleosporaceae* phylogeny presented by Pem et al. [38], placing the genus *Gibbago* in a fully supported clade related to *Exserohilum*, which are both placed far from the new genus proposed here.

***Scleromyces*** Torres-Garcia, Dania García and Gené, gen nov.

MycoBank: MB 843291

*Etymology*: Name refers to the production of only sclerotium-like structures in in vitro conditions.

Subclass classification: Dothideomycetes, Pleosporomycetidae, Pleosporales, Pleosporaceae.

*Type species*: *Scleromyces submersus* Torres-Garcia, Dania García and Gené.

*Description*: *Teleomorph* and *Anamorph* not observed. *Hyphae* septate, branched, at first cylindrical, hyaline to subhyaline, smooth- and thin-walled, becoming nodose, pale olivaceous to brown and thick-walled at irregular intervals and giving rise to multicellular dark pigmented sclerotium-like structures. *Sclerotium-like structures* immersed to erumpent, discrete to confluent, multi-celled, globose, subglobose, or irregularly shaped, olivaceous brown to dark brown, smooth- and thick-walled.

*Habitat and geographical distribution*: Aside from our freshwater sediment isolate from Spain, the environmental metadata suggest that members of *Scleromyces* would inhabit temperate climate areas (Australia and USA), colonizing primarily soils but also it can be found associated with plant material (roots and shoots) (Figure 2, Table 2).

***Scleromyces******submersus*** Torres-Garcia, Dania García and Gené, sp. nov. Figure 4.

MycoBank: MB 843292

*Etymology*: The name refers to the substrate, fluvial sediments, from which the type species was collected. 

*Type*: Spain, Catalonia, La Noguera, Camarasa, Riu Segre, 41°53′07.4″ N 0°52′45.2″ E, fluvial sediments, December 2019, *D. Torres-Garcia* and *J. Gené* (**holotype** CBS H-24944, cultures ex-type FMR 18289, CBS 149025).

*Description*: *Mycelium* superficial and immersed. *Hyphae* septate, branched, at first cylindrical, 2.5–5.5 µm wide, hyaline to subhyaline, smooth- and thin-walled, becoming nodose at irregular intervals, with inflated cells up to 8.5–12.5 µm wide, pale olivaceous to brown and thick-walled, giving rise to multicellular dark pigmented sclerotium-like structures in all culture media tested. *Sclerotium-like structures* abundant, immersed to erumpent, discrete to confluent, composed of clusters of septate, pale olivaceous to brown, swollen and thick-walled cells, forming globose, subglobose, or irregularly shaped masses, up to 190 µm diam., olivaceous brown to dark brown and smooth-walled.

Based on alignments of the separate loci, *Sc. submersus* differs from its closest phylogenetic neighbor, *A. avenicola*, by having unique fixed alleles in each of the five loci: LSU positions 447 (G), 466 (T), 474 (T), 475 (C), 499 (G), 508 (C), 665 (T); ITS positions 51 (T), 59 (T), 60 (T), 62 (A), 63 (T), 64 (C), 67 (C), 69 (T), 77 (G), 97 (A), 99 (C), 100 (A), 300 (T), 312 (C), 333 (A), 348 (G), 374 (A), 376 (T), 379 (A), 382 (T), 383 (C), 384 (T), 386 (C), 388 (A); *gapdh* positions 48 (G), 49 (A), 56 (C), 59 (G), 60 (T), 61 (A), 62 (G), 63 (C), 65 (A), 67 (C), 68 (A), 69 (T), 70 (G), 76 (A), 79 (T), 81 (T), 85 (T), 86 (T), 91 (C), 92 (G), 115 (T), 139 (T), 145 (T), 161 (A), 163 (C), 167 (G), 169 (A), 172 (T), 173 (G), 175 (G), 178 (A), 179 (C), 182 (T), 186 (G), 187 (T), 190 (T), 194 (C), 195 (C), 196 (A), 197 (G), 198 (T), 199 (C), 201 (T), 205 (G), 207 (C), 210 (T), 211 (T), 212 (C), 214 (A), 216 (C), 217 (A), 219 (A), 221 (C), 222 (T), 223 (A), 224 (A), 226 (C), 228 (A), 229 (C), 231 (G), 233 (G), 235 (C), 236 (A), 237 (T), 238 (C), 239 (A), 242 (T), 243 (T), 244 (A), 250 (C), 253 (T), 254 (A), 255 (A), 257 (A), 258 (G), 261 (A), 271 (A), 286 (C), 289 (T), 295 (C), 313 (T), 316 (C), 325 (G), 238 (T), 331 (A), 346 (C), 358 (T), 409 (T), 433 (T), 436 (T), 487 (T), 496 (A), 511 (T), 514 (C), 541 (T), 556 (C), 565 (G), 571 (T), 577 (G), 583 (A), 586 (T), 592 (G); *rpb*2 positions 3 (G), 20 (T), 26 (A), 29 (C), 32 (T), 36 (T), 44 (T), 47 (G), 50 (T), 53 (T), 59 (G), 62 (C), 71 (C), 74 (G), 77 (G), 83 (A), 107 (C), 116 (C), 119 (T), 134 (A), 137 (C), 156 (T), 158 (G), 168 (C), 174 (C), 185 (T), 194 (T), 200 (C), 206 (G), 209 (G), 215 (T), 218 (C), 230 (A), 232 (T), 245 (G), 248 (G), 251 (A), 254 (G), 263 (T), 269 (C), 272 (A), 284 (A), 305 (T), 308 (T), 314 (C), 317 (C), 320 (G), 326 (A), 335 (G), 338 (A), 344 (A), 353 (A), 356 (A), 362 (A), 365 (G), 368 (C), 374 (A), 389 (A), 395 (C), 407 (T), 410 (C), 425 (T), 431 (T), 434 (C), 437 (G), 443 (T), 446 (T), 449 (C), 461 (G), 464 (C), 470 (C), 485 (C), 488 (C), 494 (A), 500 (T), 506 (T), 512 (T), 533 (T), 536 (A), 543 (G), 544 (C), 545 (G), 554 (G), 562 (A), 563 (C), 564 (T), 569 (T), 575 (C), 587 (G), 593 (T), 599 (G), 605 (G), 620 (C), 623 (T), 626 (C), 638 (G), 641 (T), 644 (G), 650 (T), 653 (T), 656 (T), 662 (A), 665 (A), 674 (T), 683 (A), 692 (C), 698 (C), 704 (A), 707 (A), 723 (A), 729 (C), 731 (A), 734 (C), 735 (A), 740 (C), 743 (C), 749 (C), 752 (A), 759 (A), 762 (G), 764 (T), 770 (C), 776 (T), 787 (C), 788 (C), 797 (T), 800 (C), 803 (T), 809 (A), 812 (T), 815 (T), 818 (C), 821 (T), 836 (T), 842 (A), 845 (G), 851 (T), 854 (C), 857 (G), 860 (C), 863 (T); *tef*1 positions 33 (C), 35 (C), 36 (C), 38 (A), 39 (C), 41 (A), 42 (T), 43 (C), 51 (A), 52 (C), 53 (T), 56 (C), 57 (T), 63 (T), 64 (G), 65 (G), 66 (T), 74 (A), 80 (C), 82 (T), 97 (A), 109 (T), 114 (G), 143 (G), 146 (C), 147 (T), 151 (A), 152 (A), 153 (G), 156 (C), 162 (G), 163 (A), 164 (A), 165 (A), 167 (G), 176 (C), 186 (A), 188 (C), 190 (G), 193 (C), 200 (T), 201 (C), 210 (C), 212 (T), 221 (A), 224 (C), 244 (T).

*Culture characteristics* (7d at 25 °C): Colonies remaining sterile in all media tested. On PDA, reaching 24–25 mm diam., slightly elevated at center, cottony and dull green (30E4) at center, velvety and yellowish grey (3B2) at periphery, margin somewhat entire, slightly fimbriate; reverse dark green (30F4) to greyish green (28E7) at center and greenish grey (28B2) at periphery. On PCA, reaching 17–18 mm diam., cottony, greyish green (29E4), margin irregular, filamentous; reverse dark green (30F5). On OA, reaching 31–33 mm diam., flattened, cottony, and dull green (28E4) at center, velvety and yellowish grey (3B2) towards periphery, margin irregular, filamentous; reverse greyish green (30E5) at center and yellowish grey (3B2) at periphery.

*Cardinal temperatures for growth*: minimum 5 °C, optimum 25 °C, maximum 35 °C.

*Distribution*: Australia and Spain (Figure 2, Table 2).

*Notes*: *Scleromyces submersus* only produced abundant sclerotium-like structures in all the culture media tested, which might resemble protoascomata. However, these never ripened nor was the type isolate able to produce conidia in any in vitro conditions to which it was submitted (see material and method section). Similar structures have been described in species of different pleosporacean genera, such as *Alternaria*, *Curvularia,* or *Pyrenophora* [18,33,42,62]. Some of their species have even been described as only producing these types of structures, as in the case of *A. slovaca* [42,63] or *P. pseudoerythrospila* [42]. However, these two fungi were classified and clearly distinguished from the other species of their respective genera using mostly the same phylogenic markers used to delineate *Sc. submersus*. The description of our fungus is clearly susceptible to emendation when new isolates of the species become available.

## 4. Discussion

The study of underexplored substrates can contribute to widening the knowledge of the *Pleosporaceae* diversity and, subsequently, to filling gaps in phylogenetic relationships among its taxa. In the present study, we describe two novel genera for the family, *Neostemphylium* and *Scleromyces*, sampled from Spanish freshwater sediments and cultured in vitro using the semi-selective medium DRBC. The efficacy of this medium to isolate pleosporacean fungi, such as *Alternaria*, *Bipolaris,* or *Curvularia*, was previously reported by Funnel-Harris et al. [64]. However, our study is the first to report that DRBC supplemented with benomyl can also be effective for culturing pleosporacean fungi of taxonomic interest, since all isolates of *Neostemphylium* were recovered from the latter medium. Anyway, its efficacy is well known to isolate ascomycetes from other groups, such as *Microascales* of clinical interest like *Lomentospora* or *Scedosporium* species [65,66].

The multi-locus phylogenetic analysis has been crucial for delimiting the novel fungi because of the resemblance of *Neostemphylium* to other genera, such as *Stemphylium* or *Gibbago*, and in the case of *Scleromyces* due to the absence of strictly sporulating structures. *Neostemphylium* shares with *Stemphylium* and *Gibbago* the production of phaeodictyospores from apically somewhat swollen and darkened conidiogenous cells [8,9]. However, it differs in the development of a synanamorph, which consists of blastic, cylindrical phragmoconidia, occasionally branched, that arise laterally or terminally on vegetative hyphae, and in the production of a microconidiation cycle not described in *Gibbago* or in *Stemphylium*. Similar structures to the *N. polymorphum* synanamorph have been described in the two species of the genus *Berkeleiomyces* (*Be.*), *Be. bassicola* and *Be. rouxiae*, although they were defined as septate chlamydospores [67]. Those fungi, however, belong to the microascaceous family *Ceratocystidaceae* and are phytopathogens to a wide range of plant hosts [67,68]. Conversely, the ability to produce a microconidiation cycle is known in the *Pleosporaceae*, since it has been reported in different species of *Bipolaris* and *Curvularia* [33,69], but the biological role of this state remains obscure. The microconidiation cycle observed in *N. polymorphum* is different from other genera because its mature secondary conidia resemble the primary ones (Figure 3N–Q), that is, they become dark brown dictyoconidia. Secondary conidia described in *Bipolaris* and *Curvularia* are small, globose, and usually one-celled [33,70].

Despite being limited to form sclerotium-like structures, *Scleromyces* is phylogenetically distinct from other genera in the *Pleosporaceae*, at least from those with available DNA sequence data (Figure 1). According to Hongsanan et al. [3], cultures for some accepted genera in the family (i.e., *Allonecte*, *Diademosa*, *Extrawettsteinina*, *Platysporoides, Pleoseptum*, *Prathoda,* and *Pseudoyuconia*) are not available for comparison and lack DNA sequence data for confirming their classification. However, most of them were described as associated with plant material, producing only the teleomorph and placed in the family according to their morphological features [3,4]. As mentioned before, there are other pleosporalean fungi, such as *A. slovaca* [18,63] or *P. pseudoerythrospila* [42], which only produce sclerotia or chlamydospores and have been distinguished from other members of the genus exclusively by molecular data. Another example in the *Pleosporales* is the recently described monotypic genus *Gambiomyces*, which has been delimited according to the phylogeny of LSU, ITS, *rpb*2, and *tef*1 and erected to accommodate the sterile fungus *G. profunda*, isolated from clinical specimens of a Gambian patient [71]. Examples from other fungal groups include the chaetothyrialean soil-inhabiting species *Cyphellophora chlamydospora*, which was described as producing only chlamydospores [72], and the species of the xylarialean endophytic genus *Muscodor*, which have been described as producing only sterile mycelia [73,74]. However, all of them have been clearly distinguished from their counterparts by their phylogeny, giving rise the possibility of naming relevant fungi like the species of *Muscodor*, which are important producers of volatile organic compounds with a wide range of potential applications in agriculture, medicine, and other sectors [74].

A huge number of unidentified environmental fungal sequences have been generated in the last decade by numerous metagenomic studies, with relevant information on ecology and distribution. One way to resolve their identification is currently to attempt to link them to sequences of known and well-established species [58,75,76,77,78]. In this context, following the recent studies on the phylogeny and global distribution of *Zanclospora* and *Codinaea* [58,59], we traced the novel species *N. polymorphum* and *Sc. submersus* in the GlobalFungi database [20] to explore their putative geographical distribution as well as to detect hypothetical hidden taxa related. Our results revealed that *N. polymorphum* is a more common worldwide fungus than *Sc. submersus*, since its sequences can be linked to a large set of environmental ITS1 sequences from samples collected in Australia, Europe (France and Spain), and the USA (Figure 2). This distribution is not surprising since those are the most-sampled areas given in the GlobalFungi database (7.46%, 33.55%, and 23.74%, respectively, at the time of accession). On the other hand, *Sc. submersus* could be defined as a rarer or more geographically restricted fungus because only one ITS2 environmental sequence, originating from Australia, matched this species. Although *N. polymorphum* and *Sc. submersus* were discovered in freshwater sediment samples, we might assume that they are more likely terrestrial fungi given that the majority of environmental sequences they were linked to originated from soils of different terrestrial biomes in regions with humid and temperate climates (Table 2). In fact, fungal communities that colonize terrestrial substrates like plants, soils, rocks, etc., will end up in the river sediments by lixiviation, where they can accumulate and survive under water conditions. That adds further support to river sediments as a suitable substrate for isolating a great diversity of fungi, including putative novel taxa.

Interestingly, our metabarcode analysis allowed us to detect four hidden phylotypes or “dark taxa”, defined by Lücking et al. [78] as “new lineages known from sequence data only but for which no individual voucher specimens or cultures exist”. Namely, one was related to the genus *Neostemphylium* (ITS1-ENV1) and three to *Scleromyces* (ITS1-ENV2, ITS1-ENV3, ITS2-ENV1) (Figure 2). However, none of them were represented by full-length ITS sequences as we did not find any correlation among the metadata from such phylotypes. That was in contrast to Réblová et al. [58], who obtained three whole ITS sequences among *Zanclospora* phylotypes, which were attributable to their geography and ecology data overlapping among various phylotypes found. We hope that all those “dark taxa” can be formally proposed soon following some of the options proposed by Lücking et al. [78] for naming fungi known only from environmental sequences.

Finally, our phylogenetic analysis has not only contributed to delineating two new genera but has also allowed us to confirm the taxonomic position of the monotypic genus *Asteromyces* [60] in the family *Pleosporaceae*, increasing to 26 the number of the accepted genera since its last review [3,4]. According to Mycobank, it was a genus classified in the family *Dematiaceae* (*Helotiales*) and in the Index Fungorum database as *incertae sedis*. We examined the morphology of the type species of the genus *A. cruciatus* and completed sequence data for its ex-type strain CBS 171.63 and for the reference strain CBS 536.92 (Table 1), confirming its particular features [60] and its relationship with other members of the family, such as *Paradendryphyella* and *Stemphylium* (Figure 1).

## Figures and Tables

**Figure 1 jof-08-00868-f001:**
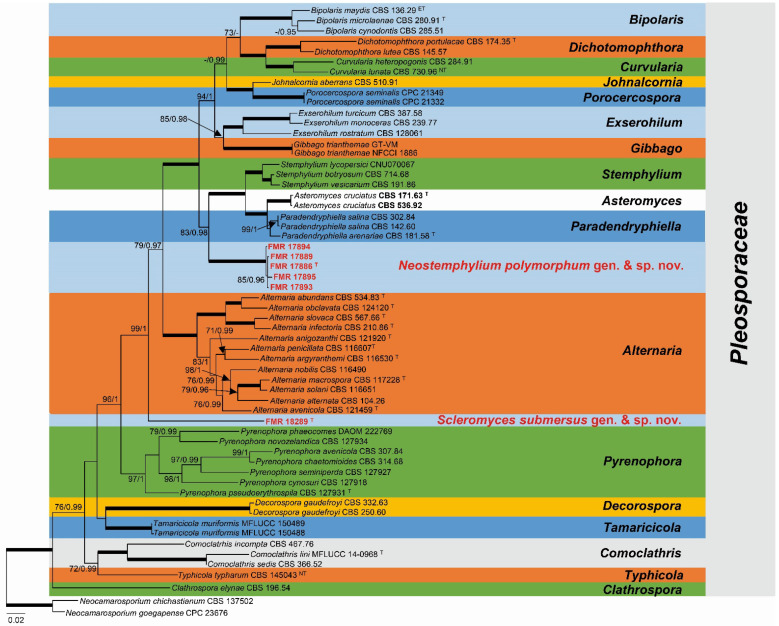
Phylogenetic analysis of the *Pleosporaceae* based on maximum likelihood analysis obtained by RAxML inferred from the combined ITS, LSU, *rpb*2, *tef*1, and *gapdh* loci. Branch lengths are proportional to phylogenetic distance. Bold branches indicate bs/pp values 100/1. Bootstrap support values/Bayesian posterior probability scores above 70%/0.95 pp are indicated on the nodes. Isolates corresponding to the new genera are shown in red. Sequences of isolates generated in this study are in bold. The tree is rooted to *Neocamarosporium chichastianum* CBS 137502 and *Neocamarosporium goegapense* CPC 23676 (*Neocamarosporiaceae*). ^T^ = ex-type strain, ^ET^ = ex-epitype strain, ^NT^ = ex-neotype strain.

**Figure 2 jof-08-00868-f002:**
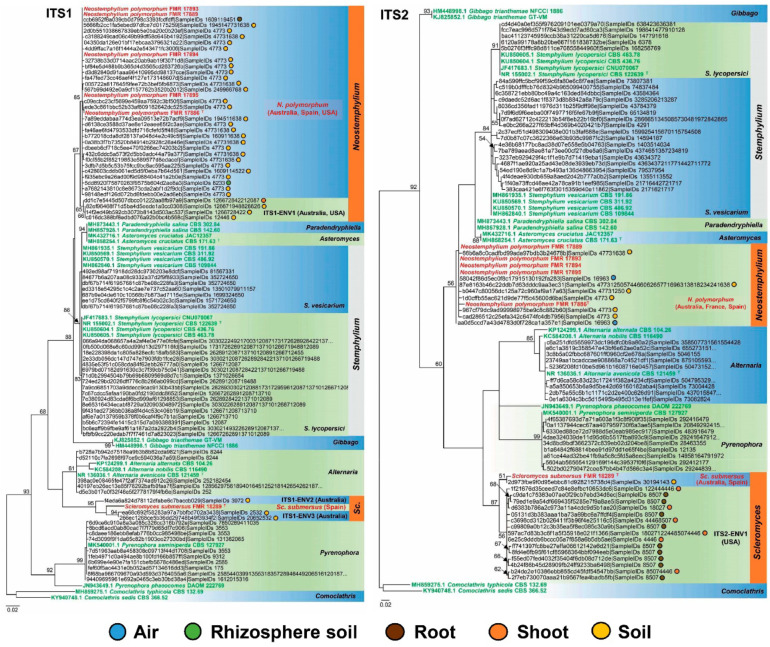
Phylogenetic relationships among *Neostemphylium* (*N*.) *polymorphum*, *Scleromyces* (*Sc*.) *submersus,* and related ITS1/ITS2 environmental sequences deposited in the GlobalFungi database. Titles of sequences contain sequence and sample codes taken from GlobalFungi. ITS1/ITS2 sequences of sediment isolates of *N. polymorphum* and *Sc. submersus* are written in red. ITS1/ITS2 sequences of known pleosporacean species are written in green. Bootstrap support values above 50% are indicated on the nodes. The trees are rooted to *Comoclathris typhicola* CBS 132.69 and *Comoclathris sedis* CBS 366.52. The trees also include data on the origin of the samples.

**Figure 3 jof-08-00868-f003:**
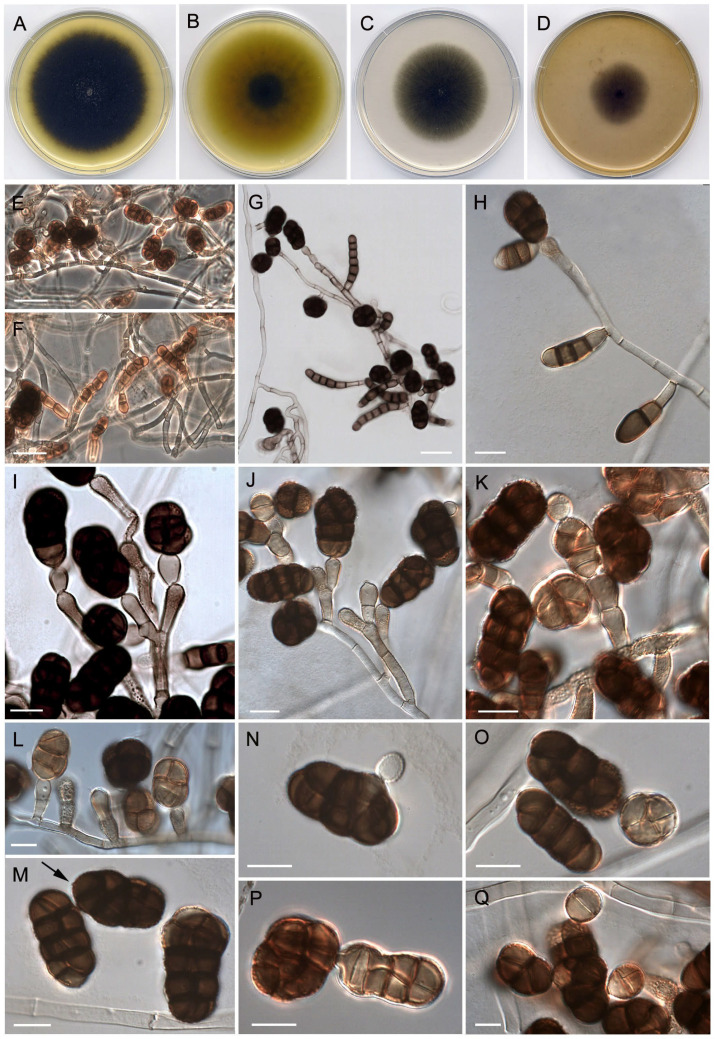
*Neostemphylium polymorphum* gen. et sp. nov. (FMR 17886). (**A**,**B**) Colonies on PDA (front and reverse). (**C**) Colony on PCA. (**D**) Colony on OA, after two weeks at 25 °C. (**E**,**I**–**L**) Conidiophores and conidia, some arranged in short acropetal chains. (**F**–**H**) Conidiophores showing phragmoconidia of the synanamorph. (**M**) Conidia with narrow cylindrical basal hylum (black arrow). (**N**–**Q**) Conidia showing different states of the microconidiation cycle. Scale bars: (**E**–**G**) = 25 μm. (**H**–**Q**) = 10 μm.

**Figure 4 jof-08-00868-f004:**
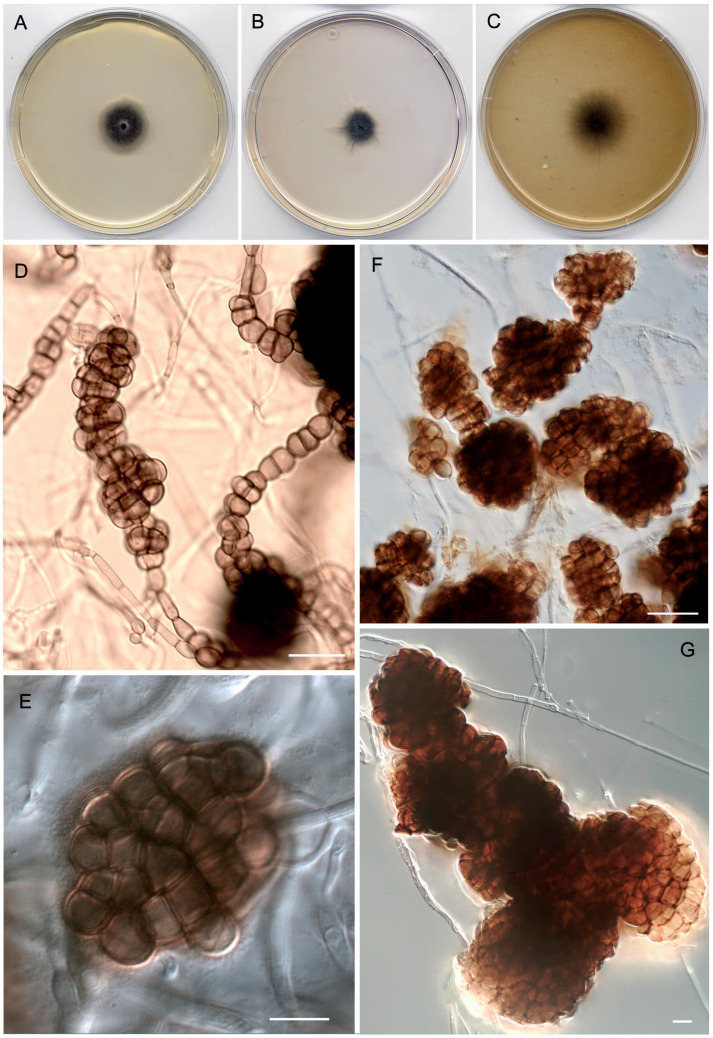
*Scleromyces submersus* gen. et sp. nov. (FMR 18289). (**A**) Colony on PDA. (**B**) Colony on PCA. (**C**) Colony on OA, after two weeks at 25 °C. (**D**–**G**) Sclerotium-like structures. Scale bars: (**D**–**G**) = 10 μm.

**Table 1 jof-08-00868-t001:** Details of the strains of the species included in the multi-locus phylogenetic analysis.

Species	Strain Number	Substrate	Country	GenBank Accession Number ^1^					Citation
				ITS	LSU	*rpb*2	*tef*1	*gapdh*	
*Alternaria abundans*	CBS 534.83 ^T^	*Fragaria stolon*	New Zealand	MH861639	MH873354	KC584448	KC584707	KC584154	[18]
*Alternaria alternata*	CBS 104.26	Unknown	Uknown	KP124299	KP124450	KP124767	KP125074	KP124156	[32]
*Alternaria anigozanthi*	CBS 121920 ^T^	*Anigozanthus* cultivar	Australia	KC584180	NG069123	KC584376	KC584635	KC584097	[18]
*Alternaria argyranthemi*	CBS 116530	*Argyranthemum sp.*	New Zealand	KC584181	KC584254	KC584378	KC584637	KC584098	[18]
*Alternaria avenicola*	CBS 121459 ^T^	*Avena sp.*	Norway	KC584183	KC584256	KC584380	KC584639	KC584100	[18]
*Alternaria infectoria*	CBS 210.86 ^T^	*Triticum aestivum*	USA	FM958526	MH873633	KC584404	KC584662	KY038017	[18]
*Alternaria macrospora*	CBS 117228 ^T^	*Gossypium barbadense*	USA	NR136045	NG069135	KC584410	KC584668	KC584124	[18]
*Alternaria nobilis*	CBS 116490 ^T^	*Dianthus caryophyllus*	New Zealand	KC584208	KC584291	KC584415	KC584673	KC584127	[18]
*Alternaria obclavata*	CBS 124120 ^T^	Air	USA	NR165505	MH874877	KC584443	KC584701	KC584149	[18]
*Alternaria penicillata*	CBS 116607 ^T^	*Papaver rhoeas*	Austria	KC584229	KC584322	KC584447	KC584706	-	[18]
*Alternaria slovaca*	CBS 567.66 ^T^	Human clinical specimen	Slovakia	KC584226	KC584319	KC584444	KC584702	KC584150	[18]
*Alternaria solani*	CBS 116651	*Solanum tuberosum*	USA	KC584217	KC584306	KC584430	KC584688	KC584139	[18]
*Asteromyces cruciatus*	CBS 171.63 ^T^	Sand of dunes	France	MH858254	MH869856	**ON703247**	**ON542234**	**ON542232**	[19]; **this study**
	CBS 536.92	Composting seaweed	USA	**ON773141**	**ON773155**	**ON703248**	**ON542235**	**ON542233**	**This study**
*Bipolaris cynodontis*	CBS 285.51	*Cynodon transvaalensis*	Kenya	MH856862	MH868380	HF934831	-	HG779081	[33]
*Bipolaris maydis*	CBS 136.29 ^ET^	*Zea mays*	Japan	MH855024	MH866491	HF934828	-	HG779086	[33]
*Bipolaris microlaenae*	CBS 280.91 ^T^	*Microlaena stipoides*	Australia	NR137073	HF934877	HF934835	-	HG779092	[33]
*Clathrospora elynae*	CBS 196.54	*Carex curvula*	Switzerland	MH857290	MH872973	KC584496	-	-	[18]
*Comoclathris incompta*	CBS 467.76	*Olaea europaea* branch	Grece	KY940770	MH871007	KC584504	-	-	[18]
*Comoclathris linis*	MFLUCC 145047 ^T^	Dead stems of *Linum* sp.	Italy	NR153904	NG058917	-	-	-	[34]
*Comoclathris sedis*	CBS 366.52	Unknown	USA	KY940748	MH871007	KT216533	-	-	[35]
*Curvularia heteropogonis*	CBS 284.91 ^T^	*Heteropogon contorus*	Australia	MH862253	LT631396	HF934821	-	HG779121	[33]
*Curvularia lunata*	CBS 730.96 ^NT^	Human lung biopsy	USA	MG722981	LT631416	HF934813	-	LT715821	[36]
*Decorospora gaudefroyi*	CBS 332.63	Unknown	France	MH858305	MH869915	-	-	-	[19]
	CBS 250.60	Unknown	UK	MH857974	MH869526	-	-	-	[19]
*Dichotomophthora lutea*	CBS 145.57 ^T^	Unknown	Unknown	MH857676	NG069497	LT990634	-	LT990663	[19]
*Dichotomophthora portulacae*	CBS 174.35 ^T^	Unknown	Unknown	NR158421	MH867137	LT990638	LT990668	-	[19]
*Didymella exigua*	CBS 183.55	*Rumex arifolius*	France	MH857436	MH871007	EU874850	-	-	[37]
*Exserohilum monoceras*	CBS 239.77	*Echinochloa colona*	Australia	LT837474	LT883405	LT852506	-	LT883547	[36]
*Exserohilum rostratum*	CBS 128061	*Zea mays*	USA	KT265240	MH877986	LT715752	-	LT715900	[36]
*Exserohilum turcicum*	CBS 387.58	*Zea mays*	USA	MH857820	LT883412	LT852514	-	LT883554	[36]
*Gibbago trianthemae*	NFCCI 1886	*Trianthema portulacastrum*	India	HM448998	MH870931	-	-	-	[19,38]
	GT-VM	*Trianthema portulacastrum*	Pakistan	KJ825852	MH870931	-	-	-	[39]
*Johnalcornia aberrans*	CBS 510.91	Unknown	Australia	MH862272	KM243286	LT715737	-	KM257056	[36]
*Neocamarosporium chichastianum*	CBS 137502	Unknown	Iran	KJ869163	MH877648	-	-	-	[40]
*Neocamarosporium goegapense*	CPC 23676	*Mesembryanthemum* sp.	South Africa	KJ869163	KJ869220	-	-	-	[40]
*Paradendriphyella arinariae*	CBS 181.58 ^T^	Unknown	France	MH857747	KC793338	DQ435065	-	-	[19]
*Paradendriphyella salina*	CBS 302.84 ^T^	*Cancer pagurus* shell	Denmark	MH873443	KC584325	KC584450	KC584709	-	[19]
	CBS 142.60	Stem of *Spartina* sp.	England	MH857928	MH869472	DQ435066	-	-	[19]
*Pheosphaeria oryzae*	CBS 110110	*Oryza sativa*	Korea	MH862850	MH871007	-	-	-	[19]
*Porocercospora seminalis*	CBS 134907	*Bouteloua dactyloides*	USA	HF934941	HF934862	HF934843	-	-	[41]
	CPC 21349	*B. dactyloides*	USA	HF934945	HF934861	HF934845	-	-	[41]
** *Neostemphylium polymorphum* **	**FMR 17886 ^T^**	**Fluvial sediment**	**Spain**	**OU195609**	**OU195892**	**OU196009**	**ON368192**	**OU195960**	**This study**
	**FMR 17889**	**Fluvial sediment**	**Spain**	**OU195610**	**OU195914**	**OU196957**	**ON368193**	**OU195977**	**This study**
	**FMR 17893**	**Fluvial sediment**	**Spain**	**OU195631**	**OU195915**	**OU197255**	**ON368194**	**OU195978**	**This study**
	**FMR 17894**	**Fluvial sediment**	**Spain**	**OU195879**	**OU195937**	**OU196956**	**ON368195**	**OU195998**	**This study**
	**FMR 17895**	**Fluvial sediment**	**Spain**	**OU195878**	**OU195936**	**OU197545**	**ON368196**	**OU195999**	**This study**
*Pyrenophora avenicola*	CBS 307.84	*Avena* seed	Sweden	MK539972	MK540042	-	-	MK540180	[42]
*Pyrenophora chaetomioides*	CBS 314.68	*Avena sativa*	Germany	MK539979	MH870853	MK540105	-	MK540187	[42]
*Pyrenophora cynosuri*	CBS 127918	Seeds of *Cynosurus*	New Zealand	MK539980	MK540047	MK540106	-	MK540188	[42]
*Pyrenophora novozelandica*	CBS 127934	Seeds of *Triticum* sp.	New Zealand	MK539997	MK540061	MK540125	-	MK540209	[42]
*Pyrenophora phaecomes*	DAOM 222769	Unknown	Unknown	JN943649	JN940093	DQ497614	-	-	[18]
*Pyrenophora pseudoerythrospila*	CBS 127931 ^T^	*Lolium* sp.	Germany	NR164465	NG066344	-	-	MK540212	[42]
*Pyrenophora seminiperda*	CBS 127927	Unknown	Unknown	MK540001	MH877966	MK540128	-	MK540213	[42]
* **Scleromyces submersus** *	**FMR 18289 ^T^**	**Fluvial sediment**	**Spain**	**OU195893**	**OU195959**	**OU197244**	**OU196982**	**OU196008**	**This study**
*Stemphylium botryosum*	CBS 714.68 ^T^	*Medicago sativa*	Canada	MH859208	MH870931	-	KC584729	MH206176	[18]
*Stemphylium lycopersici*	CNU 070067	*Capsicum annum*	Korea	JF417683	-	JF417698	JX213347	JF417693	[43]
*Stemphylium vesicarium*	CBS 191.86	*Medicago sativa*	India	MH861935	JX681120	KC584471	KC584731	-	[18]
*Tamaricicola muriformis*	MFLUCC 150488	*Tamarix* sp.	Italy	KU752187	KU561879	KU820870	-	-	[44]
	MFLUCC 150489	*Tamarix* sp.	Italy	KU752188	KU729857	-	-	-	[44]
*Typhicola typharum*	CBS 145043 ^NT^	Leaf of *Typha* sp.	Germany	MK442590	MK442530	MK442666	MK442696	-	[45]

CBS: Culture Collection of the Westerdijk Fungal Biodiversity Institute, Utrecht, the Netherlands; JAC: Culture Collection of J.A. Cooper, New Zealand; MFLUCC: Culture Collection of the Mae Fah Luang University, Chiang Rai, Thailand; NFCCI: National Fungal Culture Collection of India, Agharkar Research Institute, New Delhi, India; GT-VM: Culture Collection of V. Kumar and K.R. Aneja, Pakistan; CPC: Culture Collection of P.W. Crous, The Netherlands; FMR: Facultat de Medicina i Ciències de la Salut, Reus, Spain; DAOM: Canadian Collection of Fungal Cultures, Ottawa Research and Development Centre, Ottawa, Canada. CNU: Culture Collection of the Chungnam National University, Chungnam, South Korea; ^T^ Indicates ex-type strains; ^ET^ Indicates ex-epitype strains; ^NT^ Indicates ex-neotype strains. ^1^ ITS: Internal transcribed spacer region of the rDNA and 5.8S gene; LSU: 28S large ribosomal subunit; *rpb*2: the DNA dependent RNA polymerase II largest subunit; *tef*1: translation elongation factor 1–α; *gapdh*: glyceraldehyde-3-phosphate dehydrogenase. Novelties and sequences generated in this study are in bold.

**Table 2 jof-08-00868-t002:** The biogeography, substrate, and habitat affinity of *Neostemphylium* and *Scleromyces* environmental sequences in the GlobalFungi database.

	Abundance			Other Data ^4^			Geographical Origin ^5^			Substrate ^6^				Biomes ^7^						
Taxa	Samples ^1^	Reads ^2^	FR ^3^	MAT	MAP	pH	USA	Europe	Australia	Soil	Rhiz. Soil *	Root	Others ^8^	Wetland	Cropland	Forest	Woodland	Shrubland	Grassland	Aquatic
*Neostemphylium* *polymorphum* ITS1	3	45.56	0.079	13.0	700.2	5.8	1	1	1	1	1	1	0	1	1	0	0	0	1	0
*Neostemphylium* ITS1-ENV1	2	2.50	0.004	15.2	935.6	5.7	1	0	1	1	0	0	1	0	0	1	1	0	0	0
*Neostemphylium* *polymorphum* ITS2	2	19.88	0.064	9.7	854.4	6.0	1	1	1	1	0	0	1	0	0	1	0	0	1	0
*Scleromyces* *submersus* ITS2	1	29.00	0.140	16.4	767.0	5.8	0	0	1	1	0	0	0	0	0	0	1	0	0	0
*Scleromyces* ITS1-ENV2	1	2.00	0.012	15.0	681.0	5.8	0	0	1	1	0	0	0	0	0	0	1	0	0	0
*Scleromyces* ITS1-ENV3	1	2.00	0.012	20.1	563.0	6.7	0	0	1	1	0	0	0	0	0	0	0	1	0	0
*Scleromyces* ITS2-ENV1	2	26.87	0.044	16.8	731.4	NA ^9^	1	0	0	0	0	1	1	0	0	0	0	0	0	1

^1^ Environmental samples containing sequences belonging to a particular taxon. ^2^ Mean values of reads of a sequence belonging to a particular taxon appears across the environmental samples. ^3^ FR: Frequency of Reads, mean values of a particular taxon across samples where the taxon was found. ^4^ Average values across all samples are represented; MAT: Mean Annual Temperature (°C), MAP: Mean Annual Precipitation (mm), pH (AVG). ^5^ Presence (1) or absence (0) of a sequence of a particular taxon according to the geographical origin of environmental samples. ^6^ Presence (1) or absence (0) of sequences of a particular taxon according to the kind of substrate of a sample. ^7^ Presence (1) or absence (0) of sequences of a particular taxon according to the biome conforming the environment where samples were collected. ^8^ Other substrates in the GlobalFungi database represent air or shoots. ^9^ No data available. * Rhizosphere soil.

## Data Availability

Not applicable.

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
