# Peer review of "Two Novel Genera, Neostemphylium and Scleromyces (Pleosporaceae) from Freshwater Sediments and Their Global Biogeography"

_jof, 2022, doi:10.3390/jof8080868_

Round 1
Author Response
Manuscript ID: jof-1847094.
Firstly, we thank the reviewers for their comments and suggestions, which have allowed us to improve the content of our manuscript a lot and have given us the possibility to be published in this journal. We have practically followed all their recommendations. Responses to the list of their general remarks and suggestions are included below.
Reviewer 1 (Minor revision)
Thank you very much for the comments and corrections in the pdf file. We hope changes introduced improve the manuscript.
1.“I would like to suggest to change the title as Two Novel Genera, Neostemphylium and Scleromyces (Pleosporaceae) from…”
Response: The title has been changed according to the reviewer’s suggestion.
2.“Indicate affiliations of all authors”
Response: Author’s affiliations have been included in the new version of the ms.
Abstract
3.“Please mention unique characters of this genus, this comparison is not needed here”
Response: The comparison with other genera has been omitted, and only features of the novel genus have been mentioned.
4.“In alphabetical order”
Response: Done.
Introduction
5.“Repetition of the abstract”
Response: This first paragraph has been deleted, but to give sense to our study, part of its content has been included in the last paragraph of the introduction. We hope it will be suitable for the reviewer.
6.“What do you mean? In number of taxa? Diversity?
Response: This has been clarified with the following sentence “The Pleosporaceae is one of the largest families within the order Pleosporales (Dothideomycetes) in terms of the number of species.”
7.“Ref?”
Response: A reference support has been included.
8.“Asexual morph, please check throughout the paper”
Response: We have followed the reviewer’s suggestion and have changed "Sexual and asexual morph" to "Teleomorph and anamorph", respectively, all over the text.
9.“Pleosporaceae”
Response: Done.
10.“Which environment?”
Response: The sentence has been modified, and the types of environments are mentioned at the end of this as follows: “Members of the Pleosporaceae are widely distributed across the environment and have a wide range of lifestyles, i.e., saprophytic, endo-/epiphytic and parasitic on various hosts in terrestrial and aquatic environments”.
11.“Suggesting to combine with the above paragraph”
Response: Done.
12.“Name them here”
Response: Done. This paragraph has been modified as it has been mentioned above.
Materials and methods
13.“Alphabetical order”
Response: Done.
14.“Authors have not mentioned the steps to obtain sequence data of Asteromyces cruciatus, from where you obtained the samples? And pcr protocol etc..”
Response: Details on the strains of A. cruciatus have been added at the end of the section Sampling and isolates of the M&M in the new version. Because the unique sequences (ITS and LSU) available for A. cruciatus in the UNITE database were molecularly related to our stemphylium-like isolates, and the taxonomic information for the species was obscure and controversial in the different repositories, two strains (including the ex-type) were requested in the CBS collection for further studies. We morphologically examined both strains under the same conditions and followed the same protocol for their sequencing as for the rest of the isolates investigated in our study. The pcr protocol has been briefly explained in the new version of the ms.
15.“In capital”
Response: We do not introduce the capital letter in "potassium" to preserve the uniformity regarding the rest of the components mentioned there.
16.“Did you investigate all of the followings?”
Response: Due to a comment of the reviewer 2, we have changed the title of the Table 1, which has also clarified the doubt of the present reviewer. The title is as follows: “Details of the strains of the species included in the multi-locus phylogenetic analysis”.
17.“Cannot see in your tree”
Response: the strain JAC12357 of A. cruciatus has been deleted in the Table 1.
18.“???” referred to TreeBase
Response: We uploaded our alignments to the Zenodo webpage, which can be accessed through the URL 10.5281/zenodo.6973696.
Results
19.“Phylogenetically? Or did not exactly fit?
Response: The morphology of our isolates did not exactly fit into any known species of Stemphylium.
20.“Suggesting to add a different colour for Asteromyces strains to denote newly generated sequences”
Response: Done.
21.“Above 85 is well supported”
Response: The reviewer is right, but if he/she doesn't mind, we prefer to leave the support as it was in the original version since we would also like to show the moderate support in different nodes.
22.“No space”
Response: Checked.
23.“Image is somewhat blurred, upload high quality image”
Response: We have improved the resolution of the illustrations included in the new version of the draft. Anyway, the original version of the figures and table 2 has been uploaded to the JoF platform with the appropriate resolution.
Discussion
24.“Recitative”
Response: Changed. The section began with the sentence “The study of underexplored substrates can contribute to widening the knowledge of the Pleosporaceae diversity and, subsequently, to filling gaps in phylogenetic relationships among its taxa.”
25.“Semi-selective medium DRBC, alone or supplemented with benomyl”/ 26.“Contradictory with the above sentence”
Response: In order to recover more fungal diversity, we used DRBC with and without benomyl, and other culture media. Whereas the Scleromyces submersus strain was recovered from DRBC without that antimicrobial supplement, all Neostemphylium polymorphum strains were recovered exclusively from DRBC supplemented with benomyl. This is explained at the beginning of the section Results, and the sentences to which the reviewer refers in the Discussion have been changed to avoid confusion.
27.“Rephrase”
Response: Done.
28.“Which genera?”
Response: To clarify the sentence the names of Stemphylium and Gibbago have been added.
29.“Which family?”
Response: The name of the family has been included.
30.“Please rephrase”
Response: Done.
31.“Do they relevant here? Sordariomycetes?”
Response: These are mere examples to show that the description of novel fungi without sporulate structures has been done not exclusively in Pleosporales and to show that the obtention of cultures and its characterization allow us to identify even fungi of biotechnological interest. Then, if the reviewer doesn't mind, we would like to maintain these examples in the new version. Anyway, if the editor believes they should be omitted, we can do it.
32.“Another point to mention is the…”
Response: Deleted.
33.“Ref”
Response: We added the reference 58 (Réblová et al., 2021) as an example in which environmental unknown sequences were linked to known Zanclospora species.
34.“Furthermore…”
Response: Deleted.
35.“Rephrase”
Response: Done.
36.“What do you mean by? Did you obtained sequence data from the type?”
Response: Yes, we did. We completed sequence data from the ex-type strain of A. cruciatus (CBS 171.63) and sequenced all gene markers for the other reference strain (CBS 536.92) as explained in the ms. We requested both strains from the CBS collection not only for sequencing but also to examine their morphology since this species presents a conidiogenous apparatus completely different to the anamorphs in Pleosporaceae. We cultured this fungus under the same conditions as our isolates and sequenced the same loci following the procedure indicated in M&M.
Supplementary material
37.“Where? In the taxa table?”
Response: Yes, they are in bold and in red in the supplementary material Figure S1.
Reviewer 2 Report
The manuscript entitled " Neostemphylium and Scleromyces, two novel genera in the family Pleosporaceae from freshwater sediments and their global biogeography" presents the results of a study on the fungal diversity from freshwater sediments in Spain. The scope of the study is typical and the research conducted was done carefully using appropriate research methods. On the basis of phylogenetic and morphological analyses, the authors described two new genera. The manuscript was prepared with care and fully deserves to be published with the corrections listed in the attached PDF.

Author Response
Manuscript ID: jof-1847094.
Firstly, we thank the reviewers for their comments and suggestions, which have allowed us to improve the content of our manuscript a lot and have given us the possibility to be published in this journal. We have practically followed all their recommendations. Responses to the list of their general remarks and suggestions are included below.
Reviewer 2 (Minor revision)
We appreciate so much the comments and corrections in the pdf file. The new version of our ms has been modified according to the most of the suggestions. The list of answers to your questions are below.
Abstract
1. “Should be Ascomycota”
Response: Changed.
2. “Vage better delete”
Response: Deleted.
Introduction
3. “Please consult Kirschner, R. (2018): Sex does not sell: the argument for using the terms “anamorph” and “teleomorph” for fungi. Mycological Progress 18: 305–312. I agree that terms ‘sexual’ and ‘asexual morphs’ are widely used in newer literature but I would nevertheless strongly suggest to use teleomorph / anamorph terms because how could one claim that given taxa generate so called ‘asexual morph’ if the conidia were not tested for genetical recombinations through several subcultured generations? Heterokariosis is known in diverse anamorphic fungi, not to mention that some so called ‘asexual morphs’ are actually designed for fertilization as a spermatial producers accompanied with ascigerous stage (teleomorph) of a given species. Furthermore, there are several other ways how recombination may occur without ascigerous stage. Even so called ‘sexual morphs’ are often not immune to confusion and fundamental inaccuracies”
Response: In fact, we agree with the reviewer on the use of anamorph and teleomorph instead of sexual and asexual morphs, respectively. Therefore, this has been changed all over the text.
4. “Based on which parameters?”
Response: Morphology and phylogeny, but this paragraph has been deleted following the recommendation of the first reviewer.
5. “Didymellaceae is the largest family Please check / 6.“Please check” (regarding number of species)
Response: The reviewer is right and the sentence has been modified accordingly. However, the composition of the number of genera and species of the Pleosporaceae given on the website mentioned by the reviewer is in conflict with that of the recent taxonomic revision by Hongsanan et al (2020) and Wijayawardene et al (2020). Therefore, considering that the web still includes pleosporacean genera which have been synonymized (e.g., Embellissia or Chalastospora with Alternaria) or considered to belong to other families (e.g., Neocamarosporium in the Neocamarosporiaceae) on the basis of numerous phylogenetic studies carried out in the last decade, we prefer to follow and maintain the data published in those revisions as a more accurate representation of the current taxonomy of this group of fungi.
7. “Sexual morphs”
Response: Changed by teleomorph.
8. “Muriform”
Response: Changed
Materials and Methods
9. “Randomly?”
Response: The collection sample points were selected previously on a map. The samples were then collected in the selected sections of rivers and streams in situ randomly. This is detailed in the new version.
10. “How do you specify this?”
Response: We clarified this question by replacing "interesting fungi" with "putative novel or rare fungi".
11. “Actually, in phylogenetic trees you used collection (strain) designations, not the accession numbers!”
Response: We have changed the title of Table 1 (“Details of the strains of the species included in the multi-locus phylogenetic analysis.”) and this issue has been omitted.
12. “The database is not accessible”
Response: We uploaded in this version the alignments in Zenodo platform at: 10.5281/zenodo.6973696.
Results
13. “stemphylium-like”
Response: Changed.
14. “Curvularia strains are not monophyletic”
Response: Curvularia micropus (originally described as Helminthosporium micropus) was previously established in the genus Bipolaris (Bipolaris micropus) by Shoemaker et al. (1959). However, Hernández-Restrepo et al. (2021) placed this species in Curvularia on the basis of ITS, LSU, rpb2 and gapdh phylogenetic markers, but it was placed far from the rest of the Curvularia species analyzed. Furthermore, Dichothomophthora was not included in the analysis of Hernández-Restrepo et al. (2021), which, according to our results, seems to be closely related to C. micropus. Therefore, based on this conflict, we have removed C. micropus from our alignment, and Figure 1 has been replaced with a new analysis, leaving Curvularia as a monophyletic genus. Additionally, the Table 1 has been modified accordingly.
15. “Please use a clear version of the texts. Some are underlined with a red mark. I suggest to save your table as a PDF and use SnapShot function to copy the table”
Response: Sorry for that! We follow the instructions of the reviewer and changed the table accordingly.
16. “Please keep trying different techniques. It would be great if you could describe the teleomorph in future studies”.
Response: We will continue using different techniques and media in order to obtain and describe the teleomorph of this interesting ascomycete.
17. “Please add them to the photoplate”
Response: the presence of the synanamorphic state has been included in the legend of the Figure 3.
18. “Please check the highlights and format the references properly”
Response: All the underlined words and suggestions detailed in the references by the reviewer 2 have been checked and changed in this new version of the MS.
Reviewer 3 Report
This study presented two new genera Neostemphylium and Scleromyces from freshwater sediments. This study enriched our understanding of fungal diversity. It fits well with the scope of Journal of Fungi. The illustration of the new taxa are good. The molecular phylogeny well support the new taxa. The paper is well written. I have two question for this study.
1. Pleosporaceae is a very big family. There are many genera accepted in this family in different database or studies. Did you include all the genera in your phylogenetic tree? Or did you discuss all the related genera? I checked the NCBI and Indexfungorum, Edenia M.C. González, A.L. Anaya, Glenn, Saucedo & Hanlin is a genus of Pleosporaceae, which is not included in the phylogenetic tree of this study.
2. The new genus Scleromyces is questionable.
There is no sporulation structure observed, so there is some doubt about the establishment of a new genus.
Author Response
Manuscript ID: jof-1847094.
Firstly, we thank the reviewers for their comments and suggestions, which have allowed us to improve the content of our manuscript a lot and have given us the possibility to be published in this journal. We have practically followed all their recommendations. Responses to the list of their general remarks and suggestions are included below.
Reviewer 3 (Minor revision)
This study presented two new genera Neostemphylium and Scleromyces from freshwater sediments. This study enriched our understanding of fungal diversity. It fits well with the scope of Journal of Fungi. The illustration of the new taxa are good. The molecular phylogeny well support the new taxa. The paper is well written. I have two question for this study.
Authors thanks to reviewer 3 its comments and reflections, which are clarified in the text of the new version and commented below.
1. Pleosporaceae is a very big family. There are many genera accepted in this family in different database or studies. Did you include all the genera in your phylogenetic tree? Or did you discuss all the related genera? I checked the NCBI and Indexfungorum, Edenia M.C. González, A.L. Anaya, Glenn, Saucedo & Hanlin is a genus of Pleosporaceae, which is not included in the phylogenetic tree of this study.
Response: On the bases of morphological and/or phylogenetic investigations, Hongsanan et al. (2020) and Wijayawardene et al. (2020) accepted 23 genera in the Pleosporaceae. We followed these revisions. In our analysis were included all those with DNA sequence data available. However, as it is mentioned in the discussion of our ms, there is a set of genera (Allonecte, Diademosa, Extrawettsteinina, Platysporoides, Pleoseptum, Prathoda and Pseudoyuconia) placed in the family according exclusively to morphological features of the teleomorph from natural substrates and mostly associated with plant material (Hongsanan et al. 2020), excepting Prathoda, whose anamorphs have been related and synonymized with species of Alternaria in Index fungorum and MycoBank.
We agree with the reviewer in that there are databases with many genera accepted (an example is that given by the reviewer 2-- https://www.catalogueoflife.org/data/browse?taxonKey=625RR). However, there are genera that have been synonymized (e.g., Embellissia or Chalastospora with Alternaria) or considered to belong to other families (e.g., Neocamarosporium in the Neocamarosporiaceae or Edenia currently placed in the Phaeosphaeriaceae) on the basis of numerous phylogenetic studies carried out in the last decade. Therefore, we based our analysis of the Pleosporaceae genera on those recent fungal taxonomic revisions mentioned above as a more accurate representation of the current taxonomy of this group of fungi. We realize that our genera are susceptible to taxonomic changes in future studies, but we are just contributing to filling gaps in the family for better knowledge of its taxonomic structure and phylogenetic evolution.
2. The new genus Scleromyces is questionable. There is no sporulation structure observed, so there is some doubt about the establishment of a new genus.
Response: We are aware that Scleromyces is a genus that can be emended in future investigations, but we based our proposal not only on its phylogenetic relationships with respect to the other genera in the family with sequence data available but also on its ecology and biogeographic distribution, which also demonstrate that it is an uncommon fungus according to the Globalfungi database. Recently, Aime et al. (IMA Fungus 12:11, 2021) published a revision in which they provided a checklist for good practices in publishing the description of new taxa. There, it is mentioned the description of a novel genus in the Chaetomiaceae family (Noumeur et al., Mycol. Prog. 19:589, 2020), which is proposed from sterile mycelia, as in our case. We followed it for the description of our novel genus, Scleromyces. But other examples are in the recent literature, as it is mentioned in the discussion of our ms (i.e., Gambiomyces in the Pleoporales, or Muscodor in the Xylariales). Therefore, results obtained from our analyses and the existence of several precedents similar to our case gave us enough arguments to propose Scleromyces. We tried to induce sporulation by using different media (OA, PCA, SNA, V8, water-agar, etc) and different conditions (low or high temperatures, U.V. light, scratching the media, etc) for more than six months without success. Maybe the nature of this fungus is just to produce sclerotium-like structures.